# Changes of the aerodynamic characteristics of a flux site after an extensive windthrow

Bruna R. F. Oliveira[1], Jan J. Keizer[1], Thomas Foken[2]

[1] Earth surface processes team, Center for Environmental and Marine Studies (CESAM), Department of Environment and Planning, University of Aveiro, Aveiro, Portugal

[2] Bayreuth Center of Ecology and Environmental Research (BayCEER), University of Bayreuth, Bayreuth, Germany

*Correspondence to:* Thomas Foken (thomas.foken@uni-bayreuth.de)

**Abstract.** A maritime pine plantation in Central Portugal that has been continuously monitored using the eddy-covariance technique for carbon fluxes since a wildfire in 2017 was significantly affected by two storms during December 2019 that resulted in a large-scale windthrow. This study analyses the impacts of this windthrow on the aerodynamic characteristics of zero-plane displacement and roughness length, and, ultimately, their implications for the turbulent fluxes. The turbulent fluxes were only affected to a minor degree by the windthrow, but the footprint area of the flux tower changed markedly, so that the target area of the measurements had to be re-determined.

**Keywords:** windthrow, eddy covariance, aerodynamic characteristics, carbon dioxide fluxes

## 1 Introduction

Heterogeneous surfaces have an influence on turbulent energy and mass fluxes (Stoy et al., 2013). For example, increased fluxes have been found at forest edges (Klaassen et al., 2002). Large-eddy simulations have suggested that fluxes are at their maximum at a distance of about 10 times the canopy height from the forest edge (Kanani-Sühring and Raasch, 2015; Dupont and Brunet, 2009). An increase in flux could also be associated with an increase in stand heterogeneity (Foken et al., 2021). An intrinsic limitation of the cited studies is that they cannot find a suitable measure of heterogeneity that is correlated with turbulent fluxes.

Recent studies on the role of spatial heterogeneity in flux measurements, as referred above, either address its influence on the closure of the energy balance at the earth's surface (Mauder et al., 2020) or its relevance for footprints (Göckede et al., 2008; Chu et al., 2021). The typical aerodynamic characteristics such as zero-plane displacement and roughness length - which change following a windthrow - have been receiving little attention. The foundations for the use of aerodynamic characteristics stem from aerodynamic studies in wind tunnels, whose findings were then adopted for meteorology (Prandtl, 1932) and led to the introduction of zero-plane displacement (Paeschke, 1937). These characteristics were a significant research focus from the 1950s to 1980s, when fluxes were determined from wind and temperature profiles measured at more than 2 heights and sometimes involving elaborate procedures (Nieuwstadt, 1978; Marquardt, 1983). All relevant textbooks of today address the basic principles underpinning aerodynamic characteristics (Stull, 1988; Garratt, 1992; Arya, 2001; Foken, 2017). The fundamental problem, however, is that zero-plane displacement and roughness length can only be determined if both profile and flux measurements are available. In the past, flux measurements were typically missing so

that fluxes were estimated using complicated approximation approaches (Kader and Perepelkin, 1984). In recent times, by contrast, profile measurements are typically missing so that they are replaced by reasoned inferences on profile characteristics. In a strict sense, zero-plane displacement and roughness length can only be assumed uniform across homogeneous surfaces. Recent attempts have tried to incorporate stand structure into empirical relationships for determining zero-plane displacement and roughness length (Nakai et al., 2008; Raupach, 1994), especially by deriving stand structure from remote sensing data, based on the paper by Thom (1971). Maurer et al. (2015) carried out a large-eddy simulation to compare various approaches to estimating aerodynamic characteristics, and confirmed a near-linear relationship between canopy height and zero-plane displacement. Spatial heterogeneity and canopy structure can be addressed relatively easily by data processing, except if they change through time. Such temporal changes then lead to erroneous flux calculations. The present study wants to demonstrate this for an abrupt, dramatic change in canopy structure due to a wind throw of trees that had been killed by a wildfire some two years earlier.

The post-wildfire flux site in Vila de Rei, central Portugal (Oliveira et al., 2021) offered an opportunity to study the impacts of an abrupt change in aerodynamic characteristics, following a windthrow caused by two consecutive storms *Elsa* and *Fabien,* without an apparent concomitant change in stand heterogeneity, virtually like a laboratory experiment. Namely, the storms caused an extensive windthrow of the - dead - burnt maritime pine trees between 19 and 21 of December 2019, 28 months after the wildfire, while the tumbled trees remained on the ground afterwards (Figure 1). Still, the two storms did not throw over the - living - eucalypt trees, i.e. neither the individual eucalypt trees along the pine plantation nor those of the eucalypt plantations adjacent to it. These individual eucalypt trees expectedly have an influence on the aerodynamic characteristics (Jegede and Foken, 1999). This prompted us to examine the changes in the roughness length assuming a linear dependence of zero-plane displacement on stand height before and after windthrow. Furthermore, we decided to analyse if the changes in these two aerodynamic characteristics also affected carbon dioxide fluxes. This question, however, is difficult to answer in this particular case.  Roughly two years after the wildfire, the pine ecosystem was still in its initial phase of post-fire vegetation recovery, so that an increase in $CO_2$ uptake is to be expected between the period before and the period after the windthrow, regardless of the aerodynamic changes. Nevertheless, a change in zero-plane displacement can have a significant impact on the Obukhov-Lettau stability parameter $(z-d)/L$ ($z$: measurement height, $d$: zero-plane displacement, $L$: Obukhov length, Foken and Börngen, 2021), which, in turn, is crucial to the correction and assessment of carbon dioxide fluxes.

## 2 Materials and methods

### 2.1 Measurement site and measurements

The measurement area, instrumentation, and data processing were comprehensively described by Oliveira et al. (2021), so only the details essential for this study are given below.

The study area is located 8 km to the southwest of the Vila de Rei municipality, N39º 37' W08º 06', in a Mediterranean climate zone. The wildfire affecting the area burned 1250 ha of woodlands. The measurement site included a plateau of sedimentary sandstone deposits, located at an elevation of 250 m a.s.l. and with slopes of up to 5º over an extension of roughly 10 ha. The crowns of the bulk of the burned maritime pine (*Pinus pinaster* Ait.) were fully consumed by the fire but their trunks of approximately 8 m height remained standing. This canopy height and a zero-plane displacement of 3.8 m were used in all calculations before windthrow in Oliveira et al. (2021). While the eucalypts plantations were hardly affected

by the windthrow, the maritime pine area following the windthrow was a mixture of dead pine trunks fallen on the soil surface or on the recovering vegetation with an estimated canopy height of 2–3 m. The vegetation mainly consisted of shrubs, locally intermixed with 2–3 year old pine seedlings and a few individual, resprouting eucalypt treelets. The localized patches of burned eucalypt trees and stands had a canopy height of approximately 4 m.

In the relatively open part of the pine area, a 12m-high slim tower was installed at the end of September 2017 and equipped with an eddy-covariance system at 11.8 m, including a sonic anemometer CSAT3 (Campbell Sci. Inc.) and a LI-7500A gas analyzer (Licor Biosciences), see Figure 1. The fluxes of momentum, sensible heat, latent heat, and carbon dioxide were analyzed with the eddy-covariance method (Aubinet et al., 2012). The data of the eddy- covariance system were calculated with the Campbell Sci. Inc. EasyFlux DL software for a quick inspection in the field, while all further calculations were done with the software package TK3 (Mauder and Foken, 2015). The processing of the turbulence data is described in full detail in an extensive supplement (https://bg.copernicus.org/articles/18/285/2021/bg-18-285-2021-supplement.pdf) in Oliveira et al. (2021). Special note should be made of the use of the double rotation (Kaimal and Finnigan, 1994) and that no gap filling was applied. The analysis of the aerodynamic characteristics was done for two 9-month periods from 22 Dec. 2018 to 30 Sept. 2019 and from 22 Dec. 2019 to 30 Sept. 2020, only using data from neutral stratification conditions (6846 and 5864 half-hourly data sets for the 2018–19 and 2019–20 periods, respectively). By contrast, the analysis of the $CO_2$ flux measurements was limited to the periods from May to August in 2019 and in 2020, involving 5832 and 5759 half-hourly data sets, respectively. Only the 2020 dataset was used for assessing the influence of different aerodynamic characteristics on $CO_2$ fluxes, because the vegetation cover was markedly different in the two years.

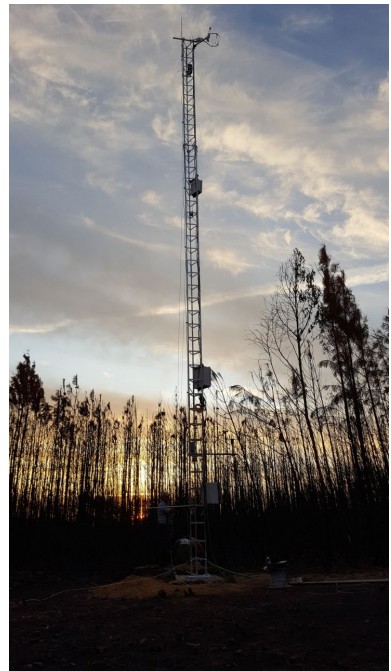 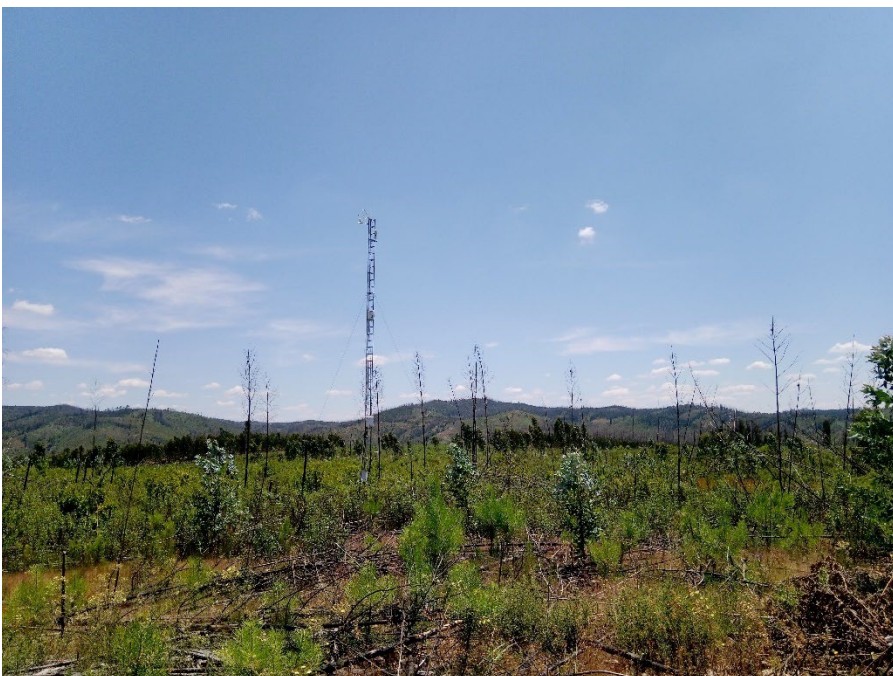

**Figure 1: 12 m slim tower with eddy-covariance system at the burnt Pinus pinaster plantation (left: Photograph: J.J. Keizer, 22 September 2017) and the plantation after the windthrow (right: Photograph: J.J. Keizer, 20 June 2020)**

**2.2. Aerodynamic characteristics**

The starting point for the analysis of the aerodynamic conditions was the logarithmic wind profile. In order to exclude influences due to the stability of the stratification, the analysis was limited to neutral cases ($-0.05 \leq z/L \leq 0.1$). The profile equation contains the measured quantities wind speed $u$ and friction velocity $u_*$ and the two unknowns of zero-plane displacement $d$ and roughness length $z_0$ (Arya, 2001; Foken, 2017; Stull, 1988)

$$\frac{u(z-d)}{u_*} = \frac{1}{\kappa} \ln \frac{z-d}{z_0} \tag{1}$$

with the von Kármán constant $\kappa = 0.4$. If wind speeds are measured at different heights, both unknowns can be determined iteratively. In the present case, however, measurements were only available at 11.8 m height, so one of the two parameters must be estimated. It is common in such cases to estimate the zero-plane displacement, in particular as equal to two-thirds of the stand height ($d = 0.666\, z_c$), as is implemented in calculation programs for eddy-covariance measurements. In reality, however, this multiplication factor varies in the range from 0.5 to 0.8, depending on stand structure and, hence, on plant development over the course of a year (Maurer et al., 2015) as well as wind speed (Marunitsch, 1971). Often the value attributed to this factor depends strongly on the experience of the observer. Oliveira et al. (2021) used as canopy height $z_c$ =7.6 m, as zero-plane displacement $d = 3.8$ m $= 0.5\, z_c$, and as roughness length $z_0 = 0.4$ m for the period before the windthrow. This determination was made because of the sparse canopy with charred trunks without leaves. For the investigations after the windthrow a canopy height $z_c = 2.7$ m and a zero-plane displacement $d = 1.8$ m $= 0.666\, z_c$ were assumed. This relationship between canopy height and zero-plane displacement corresponds to the classical approach from hydrodynamics. The application of this approach is comprehensively described in Foken (2017, Sect. 3.1.1 and 3.1.2). More recent canopy structure dependent approaches (Nakai et al., 2008; Raupach, 1994) lack input parameters for the highly disturbed surface.

The roughness length and the dimensionless wind profile $u(z-d)/u_*$ are typically used as measures of the roughness of surface and the friction on the surface. In addition, it is useful to determine the so-called integral turbulence characteristic from the standard deviation of the vertical wind velocity and friction velocity $\sigma_w/u_*$, which has a value of about 1.25 in the neutral case (Foken, 2017; Garratt, 1992) and of 1.1 for measurements close above the canopy (Finnigan et al., 2009). It can, however, attain higher values under the influence of high roughness (Foken and Leclerc, 2004).

Roughness length can be determined by two methods that are nearly independent. The first method is through Eq. (1) for a given $z-d$:

$$z_0 = \frac{z-d}{e^{\kappa\, u(z-d)/u_*}} \tag{2}$$

The second method is based on the following relation (Panofsky, 1984), with $\sigma_w/u_* = 1.25$:

$$\frac{\sigma_w}{u(z-d)} = \frac{1.25\, \kappa}{\ln\left(z-d/z_0\right)} \tag{3}$$

From Eq. (3), roughness length then follows as:

$$z_0 = \frac{z-d}{e^{\frac{1.25\,\kappa}{[\sigma_w/u(z-d)]}}} \tag{4}$$

**2.3 Influence of the changes in aerodynamic characteristics on carbon dioxide fluxes**

All relevant software packages for the calculation of the carbon dioxide fluxes use, besides measurement height, canopy height and zero-plane displacement as input parameters. These parameters are mainly needed to determine the Obukhov-Lettau stability parameter $(z - d)/L$, with the Obukhov length:

$$L = -\frac{u_*^3}{\kappa \cdot \frac{g}{T} \frac{Q_H}{\rho\, c_p}} \tag{5}$$

with the gravity accelaration $g$, the temperature $T$, the sensible heat flux $Q_H$, the air density $\rho$, and the specific heat for constant pressure $c_p$.

The Obukhov-Lettau stability parameter, in turn, is crucial for: (i) spectral correction in the high-frequency spectrum (Garratt et al., 2020; Moore, 1986); (ii) stability-dependent turbulence characteristics in quality control (Foken and Wichura, 1996); and (iii) determination of the footprint (Leclerc and Foken, 2014). Therefore, this study also addresses the implications of the windthrow-induced changes in aerodynamics for these three aspects.

The model spectra for frequency correction (e.g. Kaimal et al., 1972) are stability dependent (i), and the integral characteristics used for data quality analysis (if not limited to the neutral case as in Eq. (3)), are also stability dependent (ii), e.g. according to Panofsky et al. (1977):

$$\sigma_w/u_* = 1.3 \left(1 - 2\frac{z-d}{L}\right)^{1/3} \tag{6}$$

To determine the source area of $CO_2$-flux measurements, footprint models (iii) are used, and the input parameters are mainly wind speed, roughness length and stability. In the present case, the widely used model according to Kormann and Meixner (2001) was applied.

Further explanations of the corresponding equations and models shall be omitted since they are described comprehensively in the literature (Foken et al., 2012; Mauder et al., 2021). Furthermore, supplementary information was provided in the prior publication (Oliveira et al., 2021).

**3. Results**

**3.1. Aerodynamic characteristics**

A climatology of the wind field would require a minimum of 10 years of data, even if a period of 30 years is the standard. In other words, changes in the wind field before and after the windthrow could simply reflect the inter-annual variation. Nevertheless, it was striking that the median wind speed was higher after than before the windthrow in eight of the twelve wind sectors (Figure 2), suggesting that the windthrow provoked a generalized decrease in zero-plane displacement.

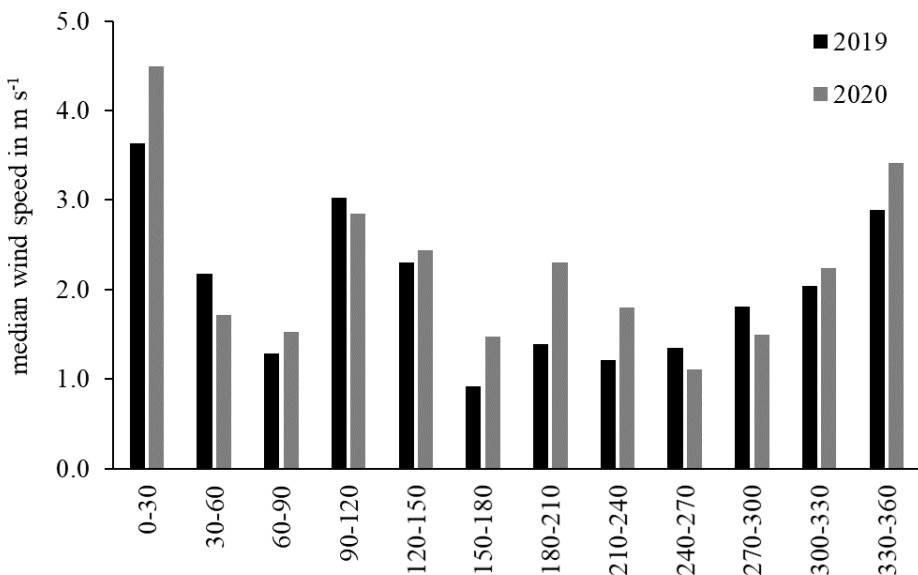

**Figure 2: Median wind speed from the 12 wind direction sectors for the periods of May to August 2019 and 2020**

According to Table 1, the ratio $u(z-d)/u_*$ before and after the windthrow changed roughly from 5–6 to 8–9, corresponding to about 60% increase. In the NE sector, the change was smaller but the ratio was already comparatively high before the windthrow. Assuming a value for $z-d$ of 8 m before and 10 m after the windthrow, the effective measuring heights were 8 m and 10 m, respectively.

The roughness length showed a decrease after the windthrow (Table 2). Especially after the windthrow, the roughness lengths agreed relatively well between the two methods. Generally, the roughness length was greatest in the SE sector.

The ratio $\sigma_w/u_*$ showed the expected values (Table 3). The values were slightly lower before than after the windthrow, in line with the smaller distance between measurement height and the top of the canopy. The NE and the W-NW sectors revealed comparatively large roughnesses both before and after the windthrow. The wind sectors highlighted in Tables 1-3 are illustrated in Figure 3.

**Table 1: The median of the ratio $u(z-d)/u_*$ before ($z-d$ = 8 m) and after ($z-d$ = 10 m) the windthrow for a nearly nine-month period (22 Dec. 2018 to 30 Sept. 2019 and 22 Dec. 2019 to 30 Sept. 2020)**

| Wind direction | 0–30 | 30–60 | 60–90 | 90–120 | 120–150 | 150–180 | 180–210 | 210–240 | 240–270 | 270–300 | 300–330 | 330–360 |
|---|---|---|---|---|---|---|---|---|---|---|---|---|
| Before windthrow | 5.25 | 6.69 | 7.38 | 5.97 | 5.15 | 5.26 | 6.48 | 5.77 | 5.79 | 6.98 | 6.26 | 5.51 |
| After windthrow | 8.47 | 8.74 | 9.52 | 7.64 | 6.60 | 8.27 | 9.19 | 9.16 | 8.71 | 10.13 | 8.51 | 8.03 |

**Table 2: The median roughness lengths $z_0$ in m before ($z - d = 8$ m) and after ($z - d = 10$ m) the windthrow for the nearly nine-month periods, computed following Eqs. (2) and (4)**

| Wind direction | 0–30 | 30–60 | 60–90 | 90–120 | 120–150 | 150–180 | 180–210 | 210–240 | 240–270 | 270–300 | 300–330 | 330–360 |
|---|---|---|---|---|---|---|---|---|---|---|---|---|
| Before windthrow | | | | | | | | | | | | |
| Eq. 2 | 0.98 | 0.55 | 0.42 | 0.73 | 1.02 | 0.98 | 0.60 | 0.80 | 0.79 | 0.49 | 0.65 | 0.88 |
| Eq. 4 | 0.89 | 0.73 | 0.38 | 0.59 | 0.86 | 0.81 | 0.48 | 0.55 | 0.65 | 0.50 | 0.61 | 0.79 |
| After windthrow | | | | | | | | | | | | |
| Eq. 2 | 0.34 | 0.30 | 0.22 | 0.47 | 0.71 | 0.37 | 0.25 | 0.26 | 0.31 | 0.17 | 0.33 | 0.40 |
| Eq. 4 | 0.27 | 0.40 | 0.26 | 0.36 | 0.57 | 0.41 | 0.23 | 0.22 | 0.28 | 0.22 | 0.34 | 0.34 |

**Table 3: The median ratio $\sigma_w/u_*$ before ($z - d = 8$ m) and after ($z - d = 10$ m) the windthrow for the nearly nine-month periods**

| Wind direction | 0–30 | 30–60 | 60–90 | 90–120 | 120–150 | 150–180 | 180–210 | 210–240 | 240–270 | 270–300 | 300–330 | 330–360 |
|---|---|---|---|---|---|---|---|---|---|---|---|---|
| Before windthrow | 1.20 | 1.39 | 1.21 | 1.15 | 1.15 | 1.15 | 1.15 | 1.07 | 1.15 | 1.26 | 1.22 | 1.19 |
| After windthrow | 1.17 | 1.36 | 1.31 | 1.15 | 1.15 | 1.30 | 1.22 | 1.21 | 1.22 | 1.33 | 1.26 | 1.19 |

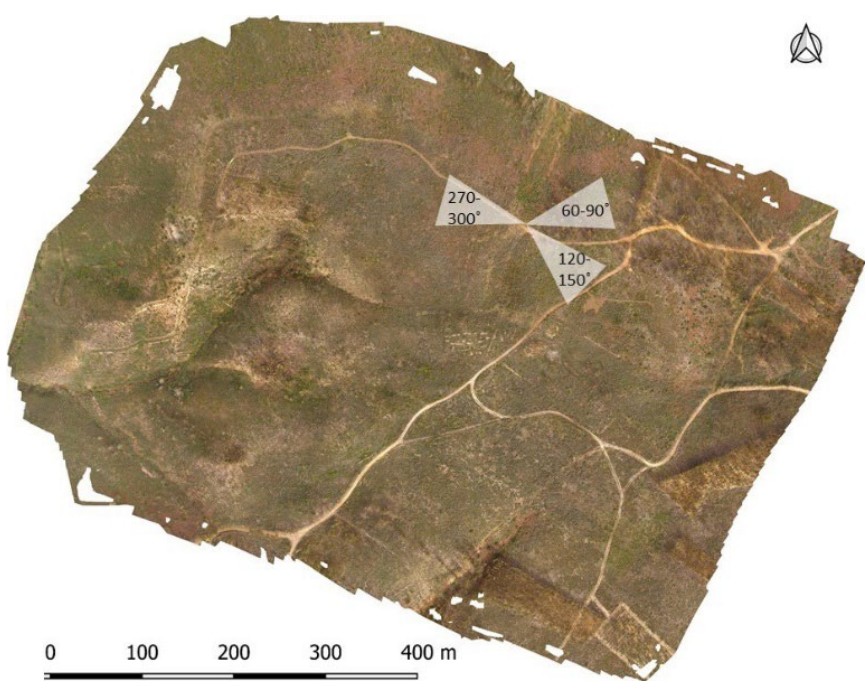

**Figure 3: Unmanned Aerial System (UAS)-based ortho-photomap with the most disturbed wind sectors. Image from 23 February 2021 by B.R.F. Oliveira.**

### 3.2 Influence on carbon dioxide fluxes

Carbon dioxide fluxes were evaluated for the months of May to August 2019 and 2020 to investigate the impacts of the windthrow-induced changes in the stability parameter. This was done without gap-filling. Of course, the comparison between before- and after-windthrow fluxes is not straightforward, because of the differences in weather conditions as well as in post-fire ecosystem recovery between the two periods. Therefore, the spectral correction and data quality analysis were only done for the 2020 data set, assuming two different zero-plane displacements.

Examination of the ratio $\sigma_w/u_*$ showed minor and not relevant differences for $z - d = 8$ m before the windthrow and $z - d = 10$ m after the windthrow.  The median differed by only 0.4 % and, hence, did not affect the quality flagging. Also, the difference with and without spectral correction was reduced, even if slightly over 1 %.

A similar result was obtained for the $CO_2$ and net ecosystem exchange (NEE) fluxes. As shown in Figure 4, the differences in NEE fluxes with and without stratification-dependent spectral correction was about 1% for the two different effective measurement heights 8 m and 10 m. The median values of the spectrally corrected fluxes differed less than 0.5 % between both measurement heights. Furthermore,  the scatter around these median values was reduced.

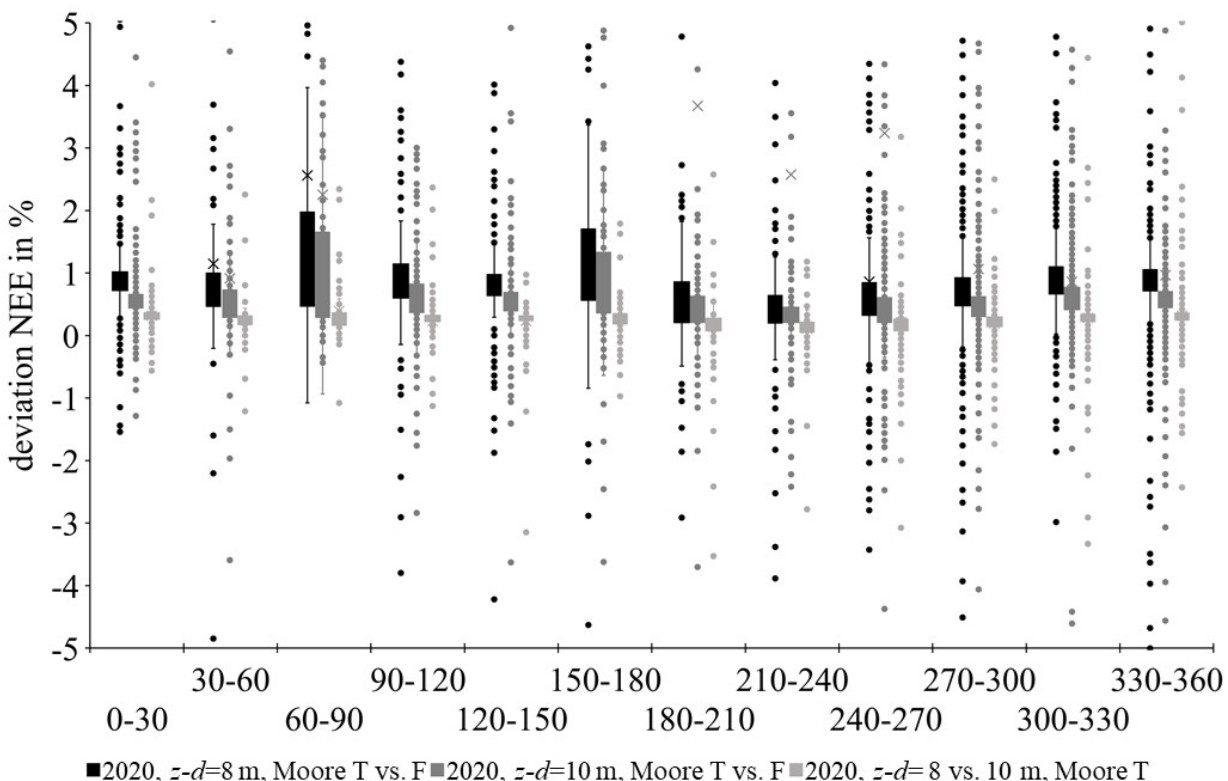

**Figure 4: Comparison of the net ecosystem excahnge (NEE) with and without spectral correction for $z - d = 8$ m (Moore T=used vs F=not used) and $z - d = 10$ m (Moore T vs F) and with spectral correction for both effective measuring heights (Moore T, $z - d = 8$ vs 10 m) for the dataset from May to September 2020**

More pronounced implications were expected with respect to the size of the source area. Therefore, the target area of burned pine woodland with down wood of charred trunks was calculated with the footprint model (Kormann and Meixner, 2001), analogous to Oliveira et al. (2021). For the 2020 period, this was done for both values of $z - d = 8$ and $z - d = 10$ and the assigned roughness lengths of $z_0 = 0.7$ m and $z_0 = 0.3$ m before and after the windthrow, respectively (Figure 5) . For comparison, Figure 5 also shows the results for the 2019 period using $z - d = 8$ m and $z_0 = 0.4$ m (Oliveira et al., 2021) as well as $z_0 = 0.7$ m (this study). The differences between the results obtained using $z - d = 8$ m and $z_0 = 0.7$ m for both periods were mainly due to the differences in wind regimes during the 2019- and 2020-periods, as shown in Figure 2. The differences between the results obtained for the 2020-period using the different aerodynamic parameter values of before and after the windthrow were relevant. The footprint was substantially larger for the post- than pre-windthrow parameter values, while the number of cases with the target area in the footprint was clearly smaller. Calculating the 2019 data with different $z_0$ values showed no substantial differences in the footprint.

The enlargement of the footprint can be prevented by reducing the measuring height. To simulate this case, Figure 5 shows the footprint for a measurement height of 9.8 m with a reduction of the wind speed by 10 % ($z\text{-}d = 8$ m, $z_0 = 0.3$ m). The change in measurement height leads to an almost identical distribution of the footprint as before the windthrow in 2019.

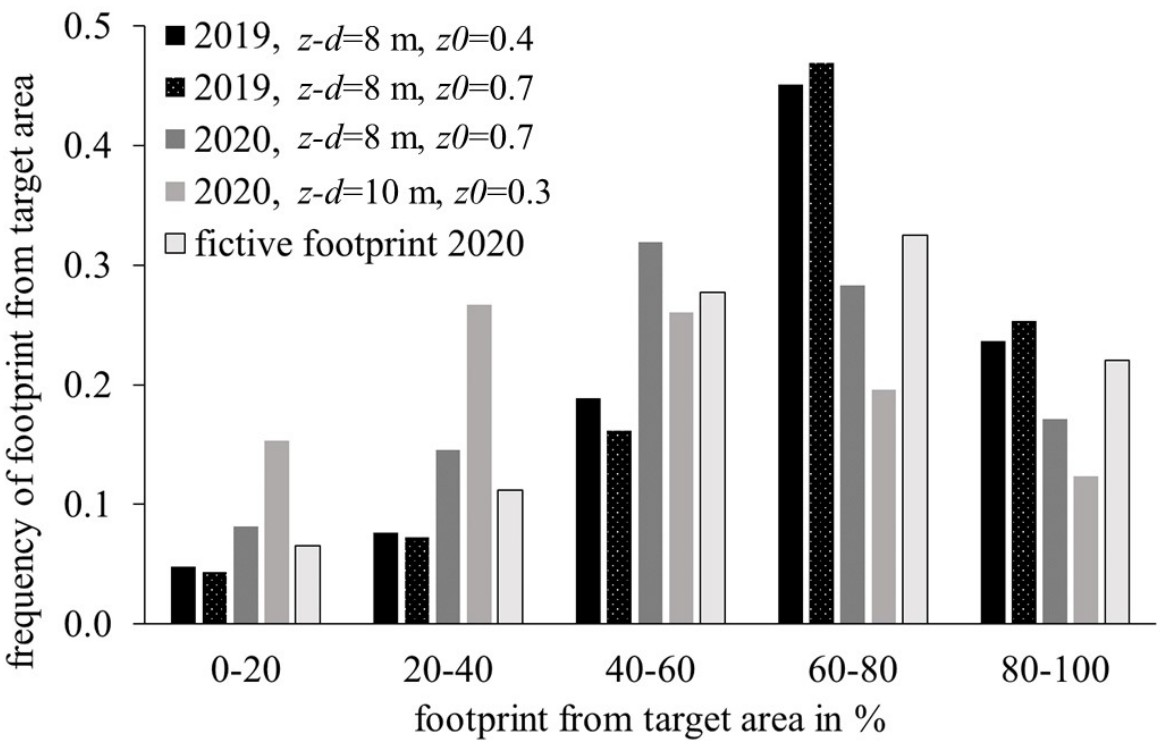

**Figure 5: Percentage of the target area (*burnt maritime pine woodland*) classes in the footprint area of the eddy-covariance measurements from May to August 2019 (before windthrow) using $z - d = 8$ m and $z_0 = 0.4$ m (following Oliveira et al., 2021) as well as using $z - d = 8$ m and $z_0 = 0.7$ m (this study) and from May to August 2020, using $z - d = 8$ m ($z_0 = 0.7$ m ) and $z - d = 10$ m( $z_0 = 0.3$ m). Furthermore, the fictive footprint for a measurement height of 9.8 m with a reduction of the wind speed by 10 % ($z\text{-}d = 8$ m, $z_0 = 0.3$ m) is shown.**

**4. Discussion**

**4.1. Aerodynamic characteristics**

Before the windthrow, the wind profile was lifted up by the displacement height. After the windthrow, the roughness of the
pine stands was determined by the vegetation that had recovered after the fire (mainly consisting of shrubs, locally intermixed
with 2–3 year old pine seedlings) and the dead pine trunks that had fallen on top of this vegetation or on the soil surface.
The two determination methods produced consistent results. Worth stressing is that the two methods (Eq. 2 and 4) are not
completely independent, because they can be transformed into each other.

The 120°-150° sector had a comparatively high roughness both before and after the windthrow. This was probably due to
the greater slope angle or the influence of the tower. By contrast, the 30°–60° and 270°–300° sectors were affected by
additional mechanical turbulence, as was also found by Oliveira et al. (2021) for the first post-fire year. This is illustrated in
Figure 6, using Eq. (1), and shows the dimensionless wind profile as a function of $z - d$ for different roughness lengths. Also
the median values for the 30° sectors were included in the plot. Because of the physical correlation between all quantities,
the plot cannot be used to determine the zero-plane displacement (as referred earlier) but it does indicate the sectors where
the assumed parameter values seem appropriate.

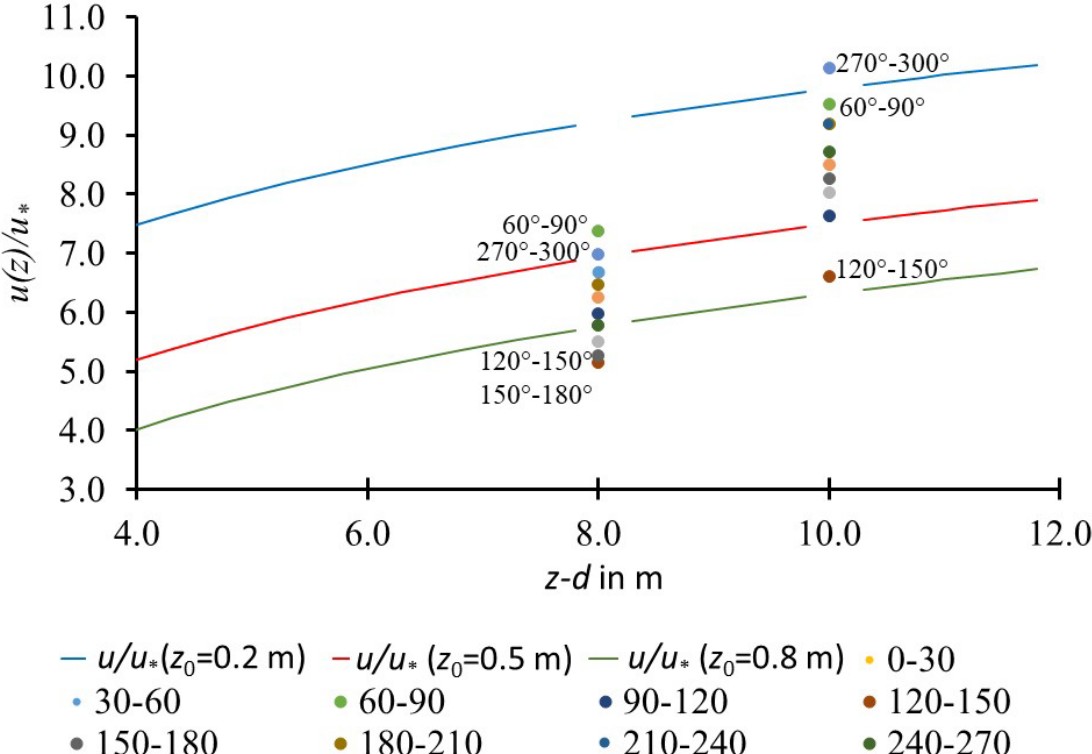

**Figure 6: Normalized wind profile in dependence on the displacement height given as $z - d$ ($z = 11.8$ m) and the
roughness length $z_0$. Median values for wind sectors are plotted for $z - d = 8$ m (before the wind break) and $z - d =
10$ m (after the wind break).**

The ratio $\sigma_w/u_*$ was largely constant at neutral stratification and only revealed a small stability dependence. This parameter can be used to detect obstacles at larger distances (Foken and Leclerc, 2004) or near the anemometer, for example. Even single standing trees can generate noticeable mechanical turbulence (Jegede and Foken, 1999). In the 60°–90° sector, the mechanical turbulence was probably due to eucalypt trees, whose crowns were quickly re-established after the wildfire by resprouting. The increased values in the 270°–330° sector can only be explained by slope parallel flow, see Oliveira et al. (2021), Figure 1.

Evaluation of both dimensionless parameters, together with an assessment of terrain and post-fire vegetation recovery, suggested that the assumptions in (Oliveira et al., 2021) of $z - d = 8$ m and $z_0 = 0.4$ m were adequate for most wind sectors for the post-fire and pre-windthrow period. However, a value for $z_0 = 0.7$ m would have more appropriate for the calculation of the footprint. Likewise, the assumptions of $z - d = 10$ m and $z_0 = 0.3$ m were appropriate for most wind sectors for the post-windthrow period. These assumptions, however, were less approriate for three sectors highlighted in Table 3, most notably after the windthrow, where no increase in the wind speed (Figure 2) could be detected. The greater variability in roughness after the windthrow can be explained by the higher wind speeds which, in turn, were due to the increase in effective measurement height resulting from the larger distance to the canopy height.

Before the windthrow, a very low roughness height of $z_0 = 0.4$ m $= 0.05 \, z_c$ was assumed because of the very wind-permeable nature of the martime stands which, in turn, reflected the fact that the fire had consumed the complete crowns of the bulk of the pine trees. This very low roughness height could not be confirmed by either the calculations using Eqs. 2 and 4 or Figure 6. The obtained value of $z_0 = 0.7$ m agreed with the simple relationship of $z_0 = 0.1 \, z_c$ (Foken, 2017; Monteith and Unsworth, 2013) but not that of $z_0 = 0.2 \, z_c$ (Kaimal and Finnigan, 1994). The same applied, mutatis mutandis, after the windthrow. By contrast, the assumed values for the ratio $d/z_c$ of 0.5 before the windthrow and 0.666 after the windthrow were confirmed by the observations. These findings further confirmed a linear relationship between $d$ and $z_c$, in line what was found using approaches that explicitly consider parameters describing stand structure (Maurer et al., 2015).

## 4.2 Influence on Carbon-dioxide fluxes

The investigations carried out clearly showed that due to the different values of the effective measurement height $z - d$ before and after the wind break in the determination of the Obukhov-Lettau stability parameter, no influence on the quality flags and the measurement of the net ecosytem exchange could be detected that would have even come close to the typical error range.

By contrast, however, the footprint area increased markedly due to the change in roughness and possibly also wind speed. The difference would probably not have been as large if a slightly better value of $z_0 = 0.7$ m had been assumed before the windthrow. The increase in footprint area also implied a decrease of the target areas in the footprint. In case target areas differ markedly from non-target areas in terms of carbon dioxide fluxes, the change in aerodynamic conditions would substantially affect flux measurements. At the present study site, this is probably not the case, as the non-target areas mainly differ from the target areas by their greater slope angles and not the burned forest. A reduction of the footprint through a reduction of the measurement height is usually not possible with long-term measurement programmes with a permanently installed mast, because reduction of the measurement height will then typically produce flow distortion problems due to the mast. Hydraulic lifted masts are hardly ever used in such programmes (Kolle et al., 2021).

## 5. Conclusions

The windthrow at the end of 2019 had a significant impact on the aerodynamics of the study area. The present analysis addressed dimensionless turbulence characteristics and focused on the parameters of roughness length and zero-plane displacement. Since both parameters are not independent, either the zero-plane displacement or the roughness length must be specified a priori. For the first post-fire year, Oliveira et al. (2021) selected a $d = 3.8$ m $= 0.5$ $z_c$ $(z - d = 8$ m), justified by the open character of the burnt pine stand due to the complete consumption of most pine crowns by the wildfire. The authors' assumption of $z_0 = 0.4$ m, however, should be revised to $z_0 = 0.7$ m. A not very precise choice of roughness length did not have a marked effect on the footprint, as the 2019 calculations showed.

The initial assumption of this study of $z - d = 10$ m and a roughness length of $z_0 = 0.3$ m for the post-windthrow period continues to seem reasonable as well. This implied that the windthrow drastically changed aerodynamic site conditions. The increase in $\sigma_w/u_*$ provided a clear indication that the distance between measurement height and canopy height had increased significantly after the windthrow.

The present study confirmed the disturbances in specific wind sectors signalled by Oliveira et al. (2021). The disturbances in the NE and NW sectors could be assigned to terrain characteristics. According to Figure 6, a change in zero-plane displacement in this sector would not result in an improvement. If coordinate rotation is performed by means of double rotation (Kaimal and Finnigan, 1994), the problem is hardly relevant for the measurements; however, if the authors would have done it by means of planar fit rotation (Wilczak et al., 2001), the disturbed sector would have to be rotated separately.

The change in aerodynamic conditions due to the windthrow did not have marked impacts on the calculation of the carbon dioxide fluxes, but it did substantially increase the footprint area. In the present case, this increase in footprint area implied the inclusion of sloping terrain but with essentially the same pre- and post-fire vegetation cover as the realtively flat target area, so that the implications are expected to be minor. The present windthrow occurred at a relative early stage of post-fire ecosystem recovery, so that a direct comparison of pre- and post-windthrow carbon dioxide fluxes was considered unwarranted.

Based on the investigation carried out, we generally recommend determining the effective measurement height and the roughness length as precisely as possible when aerodynamic conditions change in order to be able to determine changes in the footprint area. Since areas outside the target area may have an influence on the fluxes, the quality assessment of the measurement area must be carried out again, taking the footprint into account (Foken et al., 2012; Mauder et al., 2021). Alternatively, the measurement height could be adjusted so that the footprint remains almost identical.

*Code and data availability.* The program for the calculation of the eddy-covariance data is available (Mauder and Foken, 2015). The daily $CO_2$ flux data are available on Oliveira et al. (2020), other data on request from the first author (bruna.oliveira@ua.pt).

*Authors contribution.* BRFO was the responsible for ModelEco's project funding and management, and responsible scientist for this Post-doctoral study including fieldwork, data analysis ($CO_2$-part), and preparing the structure of the paper and of several of its sections; JJK was responsible for FIRE-C-BUDs' project funding and management, and data analysis (wind-

part), and revised the paper. TF was the scientific adviser of the project, mainly for the flux part and special aerodynamic analysis, and supported the writing of the paper. The final version of the paper was prepared by all authors.

*Competing interests*. The authors declare that they have no conflict of interest.

*Acknowledgements*. This work was supported by the projects ModelEco (PTDC/ASP-SIL/3504/2020), funded by national funds (OE) through FCT/MCTES, and FIRE-C-BUDs (PTDC/AGR-FOR/4143/2014 - POCI-01-0145-FEDER-016780),
funded by the FCT/MEC with co-funding by the FEDER, within the PT2020 Partnership Agreement and Compete 2020. Thanks are due to FCT/MCTES for the financial support to CESAM (UIDP/50017/2020+UIDB/50017/2020), through national funds. This publication was funded by the German Research Foundation (DFG) and the University of Bayreuth within the funding program Open Access Publishing.

*Edited by:*

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
