# Peer review of "Changes of the aerodynamic characteristics of a flux site after an extensive windthrow"

_Biogeosciences, 2022_

## Author Comment (AC1)

Answer to Reviewer 1 (answer in italic)

General comments:

The manuscript fits the scope of BG but is rather a site characterization after a sudden change in vegetation. It confirms what would be expected after the windthrow, quantifies the resulting changes, and emphasizes the relevance of properly determining aerodynamic characteristics but lacks substantial new findings.
The scientific approach is valid, however, it should be described in more detail and linked more closely to previous research findings.
Overall, the manuscript is well structured and the scientific results are presented clearly and concisely.

*We thank the reviewer for the helpful comments and will answer and consider them in detail below. We agree with the reviewer that our findings do not involve fundamentally new methods but it's precisely for that reason that we decided to submit this paper to BG and not to a different, methodology-oriented journal. Our paper targets the EC users among BG's audience who are interested in the impacts and implications of sudden structural changes in the ecosystems that they are monitoring, e.g. as a result of logging or, in the present case, windthrow. In our view, recent EC studies have paid little attention to possible aerodynamic issues in their $CO_2$ flux measurements. The bulk of the fundamental work on this topic dates back several decades, as we will illustrate by including further references to previous research findings.*

Specific comments:
(1) The paper would benefit from including more previous research findings in both the introduction and the discussion.

*We will add the following paragraph to the introduction:* **"Recent studies on the role of spatial heterogeneity in flux measurements, as referred above, either address its influence on the closure of the energy balance at the earth's surface (Mauder et al., 2020) or its relevance for footprints (Göckede et al., 2008;Chu et al., 2021). The typical aerodynamic characteristics such as zero-plane displacement and roughness length - which change following a windthrow - have been receiving little attention. The foundations for the use of aerodynamic characteristics stem from aerodynamic studies in wind tunnels, whose findings were then adopted for meteorology (Prandtl, 1932) and led to the introduction of zero-plane displacement (Paeschke, 1937). These characteristics were a significant research focus in the 1950s to 1980s, when fluxes were determined from wind and temperature profiles measured at more than 2 heights and sometimes involving elaborate procedures (Nieuwstadt, 1978;Marquardt, 1983). All relevant textbooks of today address the basic principles underpinning aerodynamic characteristics (Stull, 1988;Garratt, 1992;Arya, 2001;Foken, 2017). The fundamental problem, however, is that zero-plane displacement and roughness height can only be determined if both profile and flux measurements are available. In the past, flux measurements were typically missing so that fluxes were estimated using complicated approximation approaches (Kader and Perepelkin,**

*1984). In recent times, by contrast, profile measurements are typically missing so that they are replaced by reasoned inferences on profile characteristics. In a strict sense, zero-plane displacement and roughness height can only be assumed uniform across homogeneous surfaces. Recent attempts have tried to incorporate stand structure into empirical relationships for determining zero-plane displacement and roughness height (Nakai et al., 2008;Raupach, 1994), especially by deriving stand structure from remote sensing data, based on the paper by Thom (1971). Maurer et al. (2015) carried out a large-eddy simulation to compare various approaches to estimating aerodynamic characteristics, and confirmed a near-linear relationship between canopy height and zero-plane displacement."*

(2) In lines 21-26 you briefly discuss the influence of surface heterogeneity on turbulent fluxes. I don't see the relevance to your research since the area you're investigating is mostly homogeneous, at least concerning the vegetation, and you also state that, in this specific case, the windthrow doesn't change the heterogeneity or produce forest edge effects.

*We agree with the reviewer that the transition from lines 21-26 to lines 27 ff is not entirely conclusive and we have therefore added a paragraph in accordance with recommendation (1). In fact, our point was that we were dealing with heterogeneity in time, with the windthrow producing an abrupt change from one homogeneous surface (forest) to another homogeneous surface (dwarf shrubland). Even so, individual trees left standing after the windthrow could have significantly modified the wind profile, as shown by (Jegede and Foken, 1999).*

(3) Line 12-13 & 31-32: You state here that, among other things, you want to investigate the effect of the windthrow on the zero-plane displacement height but later (in section 2.2; Line 77) you mention that the zero-plane displacement height is estimated based on your experience. I think it's okay to do it that way. However, then the announcement to study the influence on the zero-plane displacement height itself is misleading because actually the influence of a changed zero-plane displacement height on turbulent fluxes and footprint area is studied.

*We appreciate the reviewer's perception that our statements on line 12-13 and 31-32 may induce readers to think that we carried out profile measurements for estimating zero-plane displacement. In our defence, we included the explanations on lines 70-74 to clarify why we had to assume some value. Although the assumed value was based on the long-standing experience of the last author, it was indeed just and only a guestimate. Even so, that it was an excellent guestimate is demonstrated in Fig. 6. Of course, we will carefully rephrase lines 12-13 & 31-32, to try to avoid misconceptions of the aim of our study.*

(4) Line 28: I would expect windthrow to increase stand heterogeneity by creating patches of different tree heights and roughness lengths. Why does the windthrow do not affect the heterogeneity of the stand in this case?

*The windthrow was close to 100% in the case of the pine trees (the dominant forest cover in the footprint) but basically 0% in the case of eucalypt trees. The former was probably due to the fact that almost all pine trees were killed by the fire, on the one hand, and, on the other hand, were relatively young and, hence, did not have a well-developed root system. The latter reflects the fact that Eucalyptus globulus is very well adapted to fire, re-sprouting quickly and extensively after fire (and other disturbances).*

*We will insert the remarks in line 209 already here and write in line 28: „...an abrupt change in aerodynamic properties of the - dead - pine stands, without an **apparent** concomitant increase in stand heterogeneity, virtually like a laboratory experiment. **By contrast, the two storms did not throw over the - living - eucalypt trees, i.e. neither the individual trees in the pine stands nor those of the eucalypt – plantations. These individual eucalypt trees expectedly have an influence on the aerodynamic characteristics (Jegede and Foken, 1999)**."*

(5) Section 2.1 is difficult to understand if the reader doesn't know the previously published Oliveira et al. (2021) paper. Also, the mentioned paper does not include a description of the site after the windthrow. What happened to the burned eucalypt trees? What height do the newly grown trees have now (after the windthrow)? Are the young trees maritime pine or eucalypt trees? Is it a homogeneous forest? Are there patches of different tree species/heights? If so, what size do they roughly have?

*We agree with the reviewer that the information provided in section 2.1 is rather limited, even if lines 77 and 210 do provide further details. We will include in the text both that the Oliviera et al. (2021) work refers exclusively to conditions prior to windthrow and will add the following sentence on line 51: **"While the eucalypts plantations were hardly affected by the windthrow, the maritime pine area following the windthrow was a mixture of dead pine trunks fallen on the soil surface or on the recovering vegetation with an estimated canopy height of 2-3 m. The vegetation mainly consisted of shrubs, locally intermixed with 2–3 year old pine seedlings and a few individual, resprouting eucalypt treelets**."*

(6) Line 108-114: It would enhance the quality of this paper if you properly described how you processed the data. Could you generally state more clearly on what kind of data your analysis is based on, e.g. how many measurements remained for the analysis of aerodynamic characteristics (chapter 2.2) or what steps were applied in the processing of the $CO_2$ fluxes. The information on which months were investigated and e.g. that no gap filling was applied should be stated in the methods section, not (only) in the results section. Also, the fact that you only used 2020 measurements with different roughness lengths is an important aspect of your approach that should be explained in the method section.

*The processing of the turbulence data was described in full detail in the extensive supplementary materials of Oliveira et al.(2021). If allowed by the publisher, we would prefer include the link to these supplementary materials and not provide them again as supplementary materials to this article (https://bg.copernicus.org/articles/18/285/2021/bg-18-285-2021-supplement.pdf). At the same time, we will add the following at line 113 but leave*

*the notes in the rest of the text:* **"The analysis of the aerodynamic characteristics was done for two 9-month periods from 22 Dec. 2018 to 30 Sept. 2019 and from 22 Dec. 2019 to 30 Sept. 2020, only using data from neutral stratification conditions (6846 and 5864 half-hourly data sets for the 2018–19 and 2019–20 periods, respectively). By contrast, the analysis of the $CO_2$ flux measurements was limited to the periods from May to August in 2019 and in 2020, involving5832 and 5759 half-hourly data sets, respectively. Only the 2020 dataset was used for assessing the influence of different aerodynamic characteristics on $CO_2$ fluxes, because the vegetation cover was markedly different in the two years."**

(7) Line 177-181: This should also be mentioned in the methods section, already, or moved to the discussion.

*As suggested by the reviewer, we will move this sentence to the discussion section*

(8) Line 191: Are the fallen pine trees the key factor here? In Figure 1, it appears that a new vegetation layer has emerged since the fire, which was not affected by the windthrow and is taller than the trunks lying on the ground.

*In addition to our comment on note (5), we will add:* **"After the windthrow, the roughness of the pine stands was determined by the vegetation that had recovered after the fire (mainly consisting of shrubs, locally intermixed with 2–3 year old pine seedlings) and the dead pine trunks that had fallen on top of this vegetation."**

(9) 248-250: You should consider including this in the discussion instead of adding new information in the conclusion.

*We appreciate the reviewer's point that we are adding new information in the conclusions, but we prefer to do so because we don't believe that this information deserves a discussion and because we regard it as concluding remark for users of the planar-fit method.*

Technical comments:
Line 11: remove the comma after "2017"
Line 12: "large-scale" instead of "large-sale"
Line 103: "windthrow-induced" instead of "windthow-induced"
Line 104: "and" instead of "while"
Line 111: remove the comma after "omitted"
Line 118: add "the" before "inter-annual"
Line 154: "ecosystem" instead of "ecoystem"
Line 156: I think the word "windbreak" refers more to "protection from the wind" than to something being destroyed by wind. I suggest consistently using "windthrow" instead.
Line 157: remove the second "m"
Line 157: insert comma after "Also"
Line 170: "the" instead of "ther"

*Thank you very much for your careful review and for having spotted these mistakes. We will correct all them in the next version.*

References

[revised manuscript text omitted]

---

## Author Comment (AC2)

Answer to Reviewer 2 (answer in italic)

General comments:
This paper describes changes in aerodynamic characteristics of an eddy covariance site following a windthrow. It highlights the importance of aerodynamic characteristics for flux measurements and determining the flux footprint.

The results are presented clearly and concisely, but the paper is narrowly focused on describing the specific characteristics of this site and doesn't include much discussion of the literature or the implications of this work. Overall, the paper would benefit from adding more background and discussion to connect this work to the broader context. In addition, more details should be included about the estimation of the zero-plane displacement and the sensitivity of the results to this estimation.

*We thank the reviewer for the helpful comments. The comments are largely identical to the comments made by the first reviewer, in response to which we have added a paragraph to the introduction (see response to Reviewer 1). Regarding the sensitivity of the individual methods, we would like to refer to the LES study by Maurer et al (2015) but, at the same time, have extended somewhat the discussion section .*

Specific comments:
Lines 21-26: The introduction would benefit from more clearly explaining the background and motivation for this work. This opening paragraph is focused on the effects of heterogeneity, although this doesn't seem to be the focus of this study. I suggest expanding the introduction to include more context from previous work and explain the motivation for this study.

*See our note to the introductory text.*

Lines 31-39: Since this study simply uses an estimate of zero-plane displacement before and after windthrow, this section should be rephrased to make clear that the objectives were to study the effects of the change in zero-plane displacement and roughness on the turbulence characteristics and flux footprint. Line 31 implies that this study is examining the zero-plane displacement itself.

*In line with this comment, we have rephrased the sentence of lines 31/32 as follows:* **"This prompted us to examine the changes in the roughness parameter before and after the windthrow, under the assumption of a linear dependence of zero-plane displacement on stand height."**

Lines 49-50: The previous paragraph states that an objective of the study was to examine changes in the zero-plane displacement height, so this statement about the

zero-plane displacement height is confusing. I suggest clarifying here the time periods of data that are used in this study (before and after the windthrow), and that d=3.8 just refers to the before-windthrow calculations.

*To avoid this confusion about the zero-plane displacement height, we propose to add **"before windthrow"** on line 50, after Oliveira et al. (2021), in addition to the changes proposed to lines 31/32 under the previous comment.*

Lines 74-79: Given the focus of this paper on changes in the zero-plane displacement, this section should include some justification of how these values of the zero-plane displacement were chosen.

*We believe that the change proposed to lines 31/32 also address this comment.*

The sensitivity of the results regarding the stability parameter, turbulence characteristics, and footprint to the chosen value for zero-plane displacement (and roughness) should be examined further. How do the results change if different values of $d$ and $z_0$ are used?

*Since we did not use a synthetic but a real data set (that from 2020), we see little sense in manipulating the stability and turbulence characteristics. Furthermore, as show in Figure 4, the influence of zero-plane displacement and roughness parameters on turbulent fluxes and stratification (frequency correction) was found to be irrelevant, both before and after the windthrow. Stability influences on the calculation of fluxes are described extensively in the literature (e.g. Foken et al., 2012), so that we see little added value in extending the present work in this respect.*

For example, Line 239 states that the difference in footprint area "would probably not have been as large if a slightly better value of $z_0$=0.5 m had been assumed before the windthrow." Calculations could be repeated with this different value to show the sensitivity of the calculated footprint area to $d$ and $z_0$.

*The application of the parameter set (zero-plane displacement, roughness parameter, measurement height) to the calculation of the footprint (fraction of the target area in the footprint) was performed using the 2020 dataset for the conditions before as well as after the windthrow and with reduced measurement height. As shown in Fig. 5, the obtained result seems to be as expected. Therefore, we believe that further combinations of d and $z_0$ would not provide relevant further insights. However, we agree that it is worthwhile, we will repeat the calculation with the changed roughness length for the conditions before the wind break. If these new results are relevant, we will integrate them in Fig. 5.*

The discussion sections should include more connections to the literature and discuss the broader implications of these findings.

*The additional paragraph in the introduction provides a more extensive citing of the existing literature. We used the classical approach, which is comprehensively presented and discussed in existing handbooks (e.g. Foken, (2017, Sect. 3.1.1 and 3.1.2). The newer approaches which involve a parameterization of stand structure could not be applied in our case, either after the forest fire and after the windthrow. This because the required stand parameters were too heterogeneous at our study site or, as for example in the case of the leaf area index, could not be determined from the data collected at the site or the information available from existing sources. At the same time, we have added the following sentences to the discussion:*

**Before the windthrow, a very low roughness height of $z_0 = 0.4$ m $= 0.05\,z_c$ was assumed because of the very wind-permeable nature of the martime stands which, in turn, reflected the fact that the fire had consumed the complete crowns of the bulk of the pine trees. This very low roughness height could not be confirmed by eitherthe calculations using Eqs. 2 and 4 or Fig. 6. The obtained value of $z_0 = 0.7$ m agreed with the simple relationship of $z_0 = 0.1\,z_c$ (Foken, 2017;Monteith and Unsworth, 2013) but not that of $z_0 = 0.2\,z_c$ (Kaimal and Finnigan, 1994). The same applied, mutatis mutandis, after the windthrow. By contrast, the assumed values for the ratio $d/z_c$ of 0.5 before the windthrow and 0.666 after the windthrow were confirmed by the observations. These findings further confirmed a linear relationship between $d$ and $z_c$, in line what was found using approaches that explicitly consider parameters describing stand structure (Maurer et al., 2015).**

Line 246-250: This should be moved to the discussion section.

*This statement is intended to inform users of the planar-fit method and, in our opinion, fits better in the Conclusions than in the Discussion, including because it does not deserve discussion  We will again emphasize at an appropriate place in the text that double rotation was used in all calculations, as this has only been described in detail so far in Oliveira et al.(2021).*

Some of the data is not publicly available and is listed as "available on request." From my understanding, Biogeosciences' data policy requires data to be in a public repository.

*Changes of $z_0$ and $d$ in the flux and footprint calculations must be made in the programs that use the 20 Hz data as input data. Such high-resolution data sets are very large and are usually not published. Furthermore, we believe that the analysis of this detailed data sets would require our active involvement, because of our in-depth knowledge of the data set as well as of the ancillary data set and the study site in genera. Therefore, we provide the dataset only to interested colleagues "on request". Worth stressing is perhaps that the dataset used in Oliviera et al. (2021, Results) has already been published (Oliveira et al., 2020). At the moment, we are analysing the EC data over the first three post-fire years and, in due time, will also make this data set available in the same format as Oliveira (2020).*

Technical corrections:
Line 170: "For the 2020 period"

Line 175: "windthrow were"
Line 243: Remove second period after "reasonable as well"

*Thank you very much for your careful review and for having spotted these mistakes. We will correct all of them in the next version.*

Foken, T., Leuning, R., Oncley, S. P., Mauder, M., and Aubinet, M.: Corrections and data quality in: Eddy Covariance: A Practical Guide to Measurement and Data Analysis, edited by: Aubinet, M., Vesala, T., and Papale, D., Springer, Dordrecht, Heidelberg, London, New York, 85-131,  doi: 10.1007/978-94-007-2351-1_4, 2012.

Foken, T.: Micrometeorology, 2nd ed., Springer, Berlin, Heidelberg, 362 pp.,  doi: 10.1007/978-3-642-25440-6, 2017.

Kaimal, J. C., and Finnigan, J. J.: Atmospheric Boundary Layer Flows: Their Structure and Measurement, Oxford University Press, New York, NY, 289 pp.,  doi: 10.1093/oso/9780195062397.001.0001, 1994.

Maurer, K. D., Bohrer, G., Kenny, W. T., and Ivanov, V. Y.: Large-eddy simulations of surface roughness parameter sensitivity to canopy-structure characteristics, Biogeosci., 12, 2533-2548, doi: 10.5194/bg-12-2533-2015, 2015.

Monteith, J. L., and Unsworth, M. H.: Principles of Environmental Physics, 4th edition, Academic Press, Oxford, 401 pp.,  doi, 2013.

Oliveira, B. R. F., Keizer, J. J., and Foken, T.: Daily Carbon Dioxide fluxes measured by an eddy-covariance station in a recently burnt Mediterranean pine stand in Central Portugal, PANGAEA, doi: 10.1594/PANGAEA.921281, 2020.

Oliveira, B. R. F., Schaller, C., Keizer, J. J., and Foken, T.: Estimating immediate post-fire carbon fluxes using the eddy-covariance technique, Biogeosci., 18, 285-302, doi: 10.5194/bg-18-285-2021, 2021.

---

## Author Response (AR2)

*General comments:*

*The authors have addressed most of the reviewer comments, and the connections between this study and the current literature have been strengthened. Overall, I believe*

*this study will be a useful contribution to the literature regarding the influence of aerodynamic characteristics on flux footprints.*

*Specific comments:*

*1. The added paragraph in lines 28-46 has improved the introduction, but the transition between this paragraph and the following paragraph is still a bit unclear. In the response to Reviewer 1 comment 2, the authors state that their point was that they were dealing with heterogeneity in time; however, the first two paragraphs of the introduction are discussing spatial heterogeneity, which seems not directly in line with this study's objectives.*

*2. Line 187: "Carbon" should not be capitalized.*

*3. Line 262: Add a space between "either" and "the".*

*4. A number of the in-text citations are missing a space after the semicolon separating multiple citations.*

*5. Line 310: This sentence seems a bit out-of-place as the last sentence of the conclusions; I suggest moving it earlier.*

**Response to second review:**

Thank you for the second critical review.

1. A sentence has been added on lines 46-49 to provide a better link between the first two paragraphs and the third paragraph.

2 – 4. The manuscript has been corrected as suggested.

5. The sentence was moved to the end of the first paragraph in Sect. 5 (lines 292–293).

---

## Author Response (AR3)

The manuscript was accepted without further revisions.